# Virtual Class Enhanced Discriminative Embedding Learning

**Binghui Chen**[1]**, Weihong Deng**[1]**, Haifeng Shen**[2]
[1]Beijing University of Posts and Telecommunications
[2]AI Labs, Didi Chuxing, Beijing 100193, China
chenbinghui@bupt.edu.cn, whdeng@bupt.edu.cn, shenhaifeng@didiglobal.com

## Abstract

Recently, learning discriminative features to improve the recognition performances gradually becomes the primary goal of deep learning, and numerous remarkable works have emerged. In this paper, we propose a novel yet extremely simple method **Virtual Softmax** to enhance the discriminative property of learned features by injecting a dynamic virtual negative class into the original softmax. Injecting virtual class aims to enlarge inter-class margin and compress intra-class distribution by strengthening the decision boundary constraint. Although it seems weird to optimize with this additional virtual class, we show that our method derives from an intuitive and clear motivation, and it indeed encourages the features to be more compact and separable. This paper empirically and experimentally demonstrates the superiority of Virtual Softmax, improving the performances on a variety of object classification and face verification tasks.

## 1 Introduction

In the community of deep learning, the Softmax layer is widely adopted as a supervisor at the top of the model due to its simplicity and differentiability, such as in object classification[9–11, 39] and face recognition[29, 30, 27, 36], *etc*. While, many research works take it for granted and omit the fact that the learned features by Softmax are only separable not discriminative. Thus, the performances of deep models in many recognition tasks are limited. Moreover, there are a few research works concentrating on learning discriminative features through refining this commonly used softmax layer, e.g. L-Softmax [22] and A-Softmax[21]. However, they require an annealing-like training procedure which is controlled by human, and thus are difficult to transfer to other new tasks. To this end, we intend to propose an automatic counterpart which dedicates to learning discriminative features.

In standard softmax, the input pattern $x_i$ (with label $y_i$) is classified by evaluating the inner product between its feature vector $X_i$ and the class anchor vector $W_j$ (if not specified, the basis $b$ is removed, which does not affect the final performances of DCNN verified in [22, 21]). Since the inner product $W_j^T X_i$ can be rewritten into $\|W_j\|\|X_i\|\cos\theta_j$, where $\theta_j$ is the angle between vector $X_i$ and $W_j$, and the linear softmax classifier is mainly determined by the angle [1], thus can be regarded as an angle classifier. Moreover, as shown in [33], the feature distribution leaned by softmax is 'radial', like in Fig.1. Thus, learning the angularly discriminative feature is the bedrock of softmax-based classifier. Here, we will first

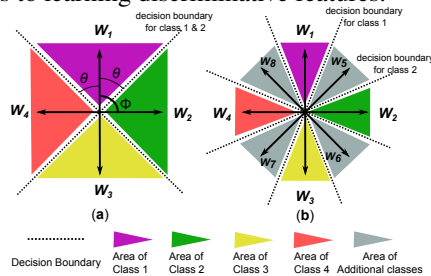

Figure 1: Illustration of angularly distributed features on 2-D space. (a) shows features learned by the original 4-way softmax. (b) shows features learned by 8-way softmax, where there are 4 additionally hand-injected negative classes. $W$ denotes the class anchor vector.

succinctly elucidate the direct motivation of our Virtual Softmax. For a 4-class classification problem, as illustrated in Fig.1.(a)[2], one can observe that when optimizing with the original 4 classes, the

decision boundary for class 1 overlaps with that for class 2, i.e. there is no inter-class angular margin. Obviously, this constraint of overlapped boundaries is not enough to ensure intra-class compactness and inter-class separability, and then contributes little to recognition performance improvement. While, in expanded 8-category constraint case as exhibited in Fig.1 .(b), the decision boundaries for class 1 and 2 have a large inter-class angular margin with each other due to the additionally injected class 5 (which doesn't take part in the final recognition evaluation) and the constrained region of class 1 is much tighter than before (other classes have the same phenomenon), yielding more compact and separable features which are highly beneficial to the original 4-class recognition. Thus, artificially injected virtual classes (e.g. $W_5 \sim W_8$) can be dedicated to encouraging larger decision margin among original classes (e.g. class $1 \sim 4$) such that produces angularly discriminative features.

Naturally motivated by this, we propose Virtual Softmax, a novel yet extremely simple technique to improve the recognition performances by training with an extra dynamic virtual class. Injecting this virtual class aims at imposing stronger and continuous constraint on decision boundaries so as to further produces the angularly discriminative features. Concretely, generalizing Softmax to Virtual Softmax gives chances of enlarging inter-class angular margin and tightening intra-class distribution, and allows the performance improvement of recognition via introducing a large angular margin. Different from L-Softmax[22] and A-Softmax[21], our work aims at extending Softmax to an automatic counterpart, which can be easily performed without or with little human interaction. Fig.2 verifies our technique where the learned features by our Virtual Softmax turn to be much more compact and well separated. And the main contributions of this paper are summarized as follows:

● We propose *Virtual Softmax* to automatically enhance the representation power of learned features by employing a virtual negative class. The learned features are much more compact and separable, illustrated in Fig.2. And to our best knowledge, it is the first work to employ additional virtual class in Softmax to optimize the feature learning and to improve the recognition performance.

● The injected virtual class derives from the natural and intuitive motivation, and intends to force a zero limit $\theta_{y_i}$ which is a much stronger constraint than softmax. Moreover, it is dynamic and adaptive, pursuing an automatic training procedure and without incurring much computational complexity and memory consumption.

● Extensive experiments have been conducted on several datasets, including MNIST [17], SVHN [23], CIFAR10/100 [16], CUB200 [35], ImageNet32[5], LFW [12] and SLLFW [6]. Finally, our Virtual Softmax has achieved competitive and appealing performances, validating its effectiveness.

## 2  Related Work

Since the training of deep model is dominated and guided by the top objective loss, and this objective function can be equipped with more semantic information, many research works force stronger representation by refining the supervisor at the top of deep model. In [29], contrastive loss is employed to enlarge the inter-class Euclidean distance and to reduce the intra-class Euclidean distance. Afterwards, it is generalized to triplet loss, the major idea is to simultaneously maximize inter-class distance and minimize intra-class distance, and it is widely applied in numerous computer vision tasks, such as in face verification [27], in person re-identification [4], in fine-grained classification [34], *etc*. Moreover, many tuple-based methods are developed and perform better on the corresponding tasks, such as Lifted loss [24], N-pair loss [28, 2], *etc*. However, these tuple-based methods constrain the feature learning with multiple instances, and thus require to elaborately manipulate the tuple mining procedure, which is much expensive in computation and is performance-sensitive. In addition, inspired by linear discriminant analysis, center loss [36] breaks away from the tuple-based idea and shows better performances. However, our Virtual Softmax differs with them in that (1) the model is optimized with only the single Virtual Softmax loss not the joint loss functions (e.g. softmax + contrastive-loss). (2) our method can also be applied to improve the classification performances, while the above methods are not as good in classification tasks. Compared to L-Softmax[22] and A-Softmax[21], this paper heads from a totally different idea that encourages a large margin among classes via injecting additionally virtual class, and dedicates to proposing an automatic method, i.e. Virtual-Softmax.

## 3  Intuition and Motivation

In this section, we will give a toy example to introduce the immediate motivation of our Virtual Softmax. Define the i-th input data $x_i$ with its corresponding label $y_i$, where $y_i \in [1 \ldots C]$ and $C$ is the class number. Define the j-th class anchor vector in softmax classifier $W_j$, where $j \in [1 \ldots C]$. The output feature of deep model is defined as $X_i$.

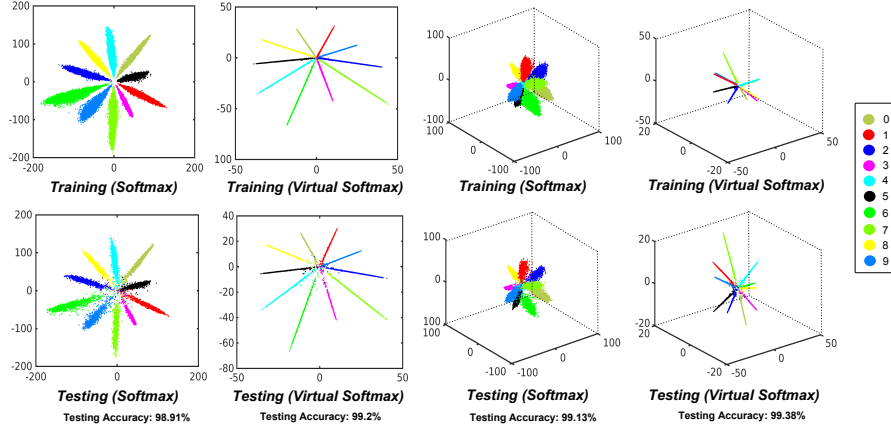

Figure 2: Visualization of the learned features optimized by original Softmax *vs.* Virtual Softmax on MNIST dataset. We provide two cases of 2-D and 3-D visualization, which are shown in the left two columns and the right two columns, respectively. From the visualization, it can be observed that our Virtual Softmax possesses a stronger power of forcing compact and separable features.

In standard softmax, the optimization objective is to minimize the following cross-entropy loss:

$$L_i = -\log \frac{e^{W_{y_i}^T X_i}}{\sum_{j=1}^{C} e^{W_j^T X_i}} \quad (1)$$

here we omit the basis $b$ and it does not affect the performance. Obviously, in original softmax optimization, it is to force $W_{y_i}^T X_i > W_j^T X_i, \forall j \neq y_i$, in order to correctly classify the input $x_i$.

Thus we can obtain the following property of softmax, which is the immediate incentive of our Virtual Softmax.

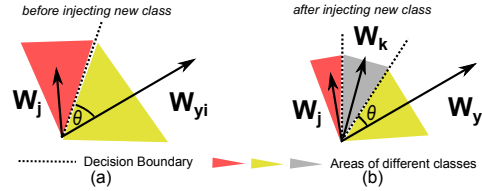

Figure 3: Toy example for non-uniform distribution case. If the class number increases, the constrained regions will be more compact than before.

**Property 1.** *As the number of class anchor vector increases, the constrained region for each class (e.g. the area of each class in Fig.1,3) becomes more and more compact, i.e. $\theta$ will gradually decreases (e.g. in Fig.1,3).*

*Proof.* For simplicity, here we consider the 2-D c-class case (n-D c-class case can be generalized as well but is more complicated.). First, assume $\|W_j\| = l, \forall j$ and they are evenly distributed in feature space, in another word, every two vectors have the same vectorial angle $\Phi = \frac{2\pi}{c}$. As stated above, the softmax is to force $W_{y_i}^T X_i > \max_{j \in c, j \neq y_i} (W_j^T X_i)$ in order to correctly classify $x_i$, i.e. $l\|X_i\| \cos \theta_{y_i} > \max_{j \in c, j \neq y_i} (l\|X_i\| \cos \theta_j) \Rightarrow \cos \theta_{y_i} > \max_{j \in c, j \neq y_i} (\cos \theta_j)$, where $\theta_j$ denotes the angle between feature vector $X_i$ and class vector $W_j$. Hence, the decision boundary for class $y_i$ is $W_{y_i}^T X = max_{j \in c, j \neq y_i} W_j^T X$, i.e. $\cos \theta_{y_i} = \max_{j \in c, j \neq y_i} (\cos \theta_j)$, we define $\theta$ as the angle between $W_{y_i}$ and the decision boundary of class $y_i$, and the solution is $\theta = \theta_{y_i} = \theta_{argmax_{j,j \neq y_i}(W_j^T X)} = \frac{\Phi}{2}$ (see Fig.1. (a) for a toy example). Considering the distribution symmetry of $W$, the angle range of the constrained region for class $y_i$ is $2\theta = \frac{2\pi}{c}$, which is inversely proportional to $c$. Thus, if the class number increases, the constrained region of each class will be more and more compact. Moreover, for the $\|W_{y_i}\| \neq \|W_j\|$ and non-uniform distribution case, the analysis is a little more complicated. Because the length of $W$ are different, the feasible angles of class $y_i$ and class $j$ are also different (see the decision boundary of the case before injecting new classes in Fig.3.(a)). Normally, the larger $\|W_{y_i}\|$ is, the larger the feasible region of the class $y_i$ is [22]. While, as illustrated in Fig.3.(b), if the class number increases, i.e. more new classes are injected to the same space, the constrained region of each class is going to be compact as well. $\square$

## 4 Virtual Softmax

Naturally motivated by the property.1, a simple way to introduce a large margin between classes for achieving intra-class compactness and inter-class separability is to constrain the features with more and more additional classes. In another word, injecting additional classes is to space the original classes, resulting in a margin between these original classes. Concretely, for a given classification task (namely that the number of categories to be classified is fixed), the additionally injected classes

(if inserted at the exact position) will introduce a new and more rigorous decision boundary for these original classes, compressing their intra-class distribution, and furthermore the originally overlapped decision boundaries of the adjacent categories will thus be forced to separate with each other, producing a margin between these decision boundaries and thus encouraging inter-class separability (see Fig.1 for a toy example). Therefore, injecting extra classes for feature supervision is a bold yet reasonable idea to enhance the discriminative property of features, and the objective function can be formulated as:

$$L_i = -\log \frac{e^{W_{y_i}^T X_i}}{\sum_{j=1}^{C+K} e^{W_j^T X_i}} = -\log \frac{e^{\|W_{y_i}\|\|X_i\|\cos\theta_{y_i}}}{\sum_{j=1}^{C+K} e^{\|W_j\|\|X_i\|\cos\theta_j}} \tag{2}$$

where $K$ is the number of extra injected classes. From Eq. 2, it can be observed that it is more possible to acquire a compact region for class $y_i$ quantified by the angle $\theta_{y_i}$, since the angle $\theta_{y_i}$ is optimized above a much larger set (C+K classes)[3].

Theoretically, the larger $K$ is, the better features are. Nevertheless, in practical cases where the available training samples are limited and the total number of classes is changeless, it is intractable to enlarge the angular margin with real and existed extra classes. Moreover, a most troublesome problem is that we can not insert the extra classes exactly between the originally adjacent categories due to the random initialization of class anchor vectors $W$ before training and the dynamic nature of parameter update during optimizing.

To address the aforementioned issues, we introduce a single and dynamic negative class into original softmax. This negative class is constructed on the basis of current training instance $x_i$. Since there is no real training data belonging to this class and it is employed only as a negative category (i.e. this class is utilized only to assist the training of original C classes and have no need to be treated as a positive class), we denote it as virtual negative class and have the following formulation of our proposed ***Virtual Softmax***:

$$L = \frac{1}{N} \sum_{i=1}^{N} L_i = -\frac{1}{N} \sum_{i=1}^{N} \log \frac{e^{W_{y_i}^T X_i}}{\sum_{j=1}^{C} e^{W_j^T X_i} + e^{W_{virt}^T X_i}} \tag{3}$$

where $W_{virt} = \frac{\|W_{y_i}\|X_i}{\|X_i\|}$ and $N$ is the training batch size. In the above equation, we instead require $W_{y_i}^T X_i \geq \max \underbrace{(W_1^T X_i \ldots W_C^T X_i, W_{virt}^T X_i)}_{C+1}$, it is a special case of Eq. 2 where $K = 1$. Replacing a large and fixed set of virtual classes (like $K$ classes in Eq. 2) with a single dynamic virtual class $W_{virt}$ incurs nearly zero extra computational cost and memory consumption compared to original softmax.

From Eq.3, one can observe that this virtual class is tactfully inserted around the class $y_i$, and particularly it is inserted at the same position with $X_i$ as illustrated in Fig.4.(a). It well matches our motivation that insert negative class between the originally adjacent classes $W_{y_i}$ and $W_j$, and this negative class will never stop pushing $X_i$ towards $W_{y_i}$ until they overlap with each other due to the dynamic nature of $X_i$ during training procedure. Moreover, from another optimization perspective, in order to correctly classify $x_i$, Virtual Softmax forces $W_{y_i}^T X_i$ to be the largest one among $(C+1)$ inner product values. Since $W_{virt}^T X_i = \|W_{y_i}\|\|X_i\|$, the only way to make $W_{y_i}^T X_i \geq \max_{j \in C+1}(W_j^T X_i) = W_{virt}^T X_i$, i.e. $\|W_{y_i}\|\|X_i\|\cos\theta_{y_i} \geq \|W_{y_i}\|\|X_i\|$, is to optimize $\theta_{y_i} = 0$, however, the original softmax is to optimize $\|W_{y_i}\|\cos\theta_{y_i} \geq \max_{j \in C}(\|W_j\|\cos\theta_j)$, i.e. $\theta_{y_i} \leq \min_{j \in C}(\arccos(\frac{\|W_j\|}{\|W_{y_i}\|}\cos\theta_j))$ (briefly, if $\|W_j\|$ have the same magnitude, the original softmax is to optimize $\theta_{y_i} \leq \min_{j \in C}(\theta_j)$). Obviously, the original softmax is to make $\theta_{y_i}$ to be smaller than a certain value, while our Virtual Softmax optimizes a much more rigorous objective, i.e. zero limit $\theta_{y_i}$, aiming to produce more compact and separable features. And based on this optimization goal, the new decision boundary of class $y_i$ is overlapping with the class anchor $W_{y_i}$, which is more strict than softmax.

**Optimization**: The Virtual Softmax can be optimized with standard SGD and BP. And in backward propagation, the computation of gradients are listed as follows:

$$\frac{\partial L_i}{\partial X_i} = \frac{\sum_{j=1}^{C} e^{W_j^T X_i} W_j + e^{W_{virt}^T X_i} W_{virt}}{\sum_{j=1}^{C} e^{W_j^T X_i} + e^{W_{virt}^T X_i}} - W_{y_i}, \quad \frac{\partial L_i}{\partial W_{y_i}} = \frac{e^{W_{y_i}^T X_i} X_i + e^{W_{virt}^T X_i} \frac{\|X_i\|}{\|W_{y_i}\|} W_{y_i}}{\sum_{j=1}^{C} e^{W_j^T X_i} + e^{W_{virt}^T X_i}} - X_i \tag{4}$$

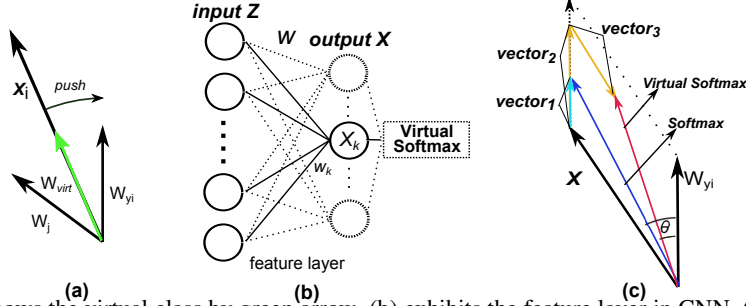

Figure 4: (a) shows the virtual class by green arrow. (b) exhibits the feature layer in CNN. (c) illustrates the feature update, the blue arrow represents $X^{'}$ obtained by original softmax and the red arrow represents $X^{'}$ obtained by Virtual Softmax.

$$\frac{\partial L_i}{\partial W_j} = \frac{e^{W_j^T X_i}}{\sum_{j=1}^{C} e^{W_j^T X_i} + e^{W_{virt}^T X_i}} X_i, where \ j \neq y_i \tag{5}$$

## 5 Discussion

Except for the above analysis that Virtual Softmax optimizes a much more rigorous objective than the original Softmax for learning discriminative features, in this section, we will give some interpretations from several other perspectives, i.e. coupling decay5.1 and feature update5.2. And we also provide the visualization of the learned features for intuitive understanding5.3.

### 5.1 Interpretation from Coupling Decay

In this paragraph, we give a macroscopic analysis from coupling decay which can be regarded as a regularization strategy.

Observing Eq.3, $L_i$ can be reformulated as: $L_i = -W_{y_i}^T X_i + \log \left( \sum_{j=1}^{C} e^{W_j^T X_i} + e^{\|W_{y_i}\| \|X_i\|} \right)$, then performing the first order Taylor Expansion for the second term, a term of $\|W_{y_i}\| \|X_i\|$ shows up. Therefore, minimizing Eq.3 is to minimize $\|W_{y_i}\| \|X_i\|$ to some extend, and it can be viewed as a coupling decay term, i.e. data-dependent weight decay and weight-dependent data decay. It regularizes the norm of both feature representation and the parameters in classifier layer such that improves the generalization ability of deep models by reducing over-fitting. Moreover, it is supported by some experimental results, e.g. the feature norm in Fig.2 is decreased by Virtual Softmax (e.g. from 100 to 50 in 2-D space), and the performance improvement over the original Softmax is increased when using a much wider network as in Sec.6.1 (e.g. in CIFAR100/100+, increasing the model width from t=4 to t=7, the performance improvement is rising, since increasing the dimensionality of parameters while keeping data set size constant calls for stronger regularization).

However, the reason of calling the above analysis a macroscopic one is that it coarsely throws away many other relative terms and only considers the single effect of $\|W_{y_i}\| \|X_i\|$, without taking into account the collaborative efforts of other terms. Thus there are some other phenomenons that cannot be explained well, e.g. why the inter-class angular margin is increased as shown in Fig.2 and why the confusion matrix of Virtual Softmax in Fig.5 shows more compact intra-class distribution and separable inter-class distribution than the original Softmax. To this end, we provide another discussion below from a relatively microscopic view that how the individual data representations of Virtual Softmax and original Softmax are constrained and formed in the feature space.

### 5.2 Interpretation From Feature Update

Here, considering the collaborative effect of all the terms in Eq.3, we give another microscopic interpretation from the perspective of feature update, which is also a strong justification of our method. In part, it reveals the reason why the learned features by Virtual Softmax are much more compact and separable than original softmax.

To simplify our analysis, we only consider the update in a linear feature layer, namely that the input to this linear feature layer is fixed. As illustrated in Fig.4.(b), the feature vector $X$ is computed as $X = w^T Z$ [4], where $Z$ is the input vector, $w$ is the weight parameters in this linear layer. We denote the $k$-th element of vector $X$ as $X_k$, the connected weight vector (i.e. the $k$-th column of $w$) as $w_k$. Thus, $X_k = w_k^T Z$, after computing the partial derivative $\frac{\partial L}{\partial w_k}$, the parameters are updated by SGD as

$w^{'}_k = w_k - \alpha \frac{\partial L}{\partial w_k}$, where $\alpha$ is the learning rate. Since $Z$ is fixed, in the next training iteration, the new feature output $X^{'}_k$ can be computed by the updated $w^{'}_k$ as:

$$X^{'}_k = (w^{'}_k)^T Z = (w_k - \alpha \frac{\partial L}{\partial w_k})^T Z = w_k^T Z - \alpha(\frac{\partial L}{\partial w_k})^T Z = X_k - \alpha(\frac{\partial X_k}{\partial w_k} \frac{\partial L}{\partial X_k})^T Z$$

$$= X_k - \alpha \frac{\partial L}{\partial X_k}(\frac{\partial X_k}{\partial w_k})^T Z = X_k - \alpha \frac{\partial L}{\partial X_k} Z^T Z = X_k - \alpha \|Z\|^2 \frac{\partial L}{\partial X_k} \tag{6}$$

thus from Eq. 6, it can be inferred that the holistic feature vector $X^{'}$ can be obtained by $X^{'} = X - \beta \frac{\partial L}{\partial X}$, where $\beta = \alpha \|Z\|^2$, implying that updating weight parameters $w$ with SGD can implicitly lead to the update of the output feature in the similar way. Based on this observation, putting the partial derivatives $\frac{\partial L}{\partial X}$ of Softmax and Virtual Softmax into Eq. 6 respectively, we can obtain the following corresponding updated features for Softmax (Eq. 7) and Virtual Softmax (Eq. 8) respectively:

$$X^{'} = X + \beta(W_{y_i} - \frac{\sum_{j=1}^{C} e^{W_j^T X} W_j}{\sum_{j=1}^{C} e^{W_j^T X}}) \tag{7}$$

$$X^{'} = X + \beta(W_{y_i} - \frac{\sum_{j=1}^{C} e^{W_j^T X} W_j + e^{W_{virt}^T X} W_{virt}}{\sum_{j=1}^{C} e^{W_j^T X} + e^{W_{virt}^T X}}) \tag{8}$$

Since the above equations are complicated enough to discuss, here for simplicity, we will give an approximate analysis. Consider a well trained feature (i.e. $e^{W_{virt}^T X} \gg e^{W_j^T X}$ and $e^{W_{y_i}^T X} \gg e^{W_j^T X}, \forall j \neq y_i$) and that the parameters in Softmax and Virtual Softmax are the same. Then, omitting the relatively smaller terms in denominators, i.e. $\sum_{j=1,j\neq y_i}^{C} e^{W_j^T X} W_j$, Eq.7, 8 can be separately approximated as:

$$X^{'} = X + \beta(W_{y_i} - \underbrace{\frac{e^{W_{y_i}^T X} W_{y_i}}{\sum_{j=1}^{C} e^{W_j^T X}}}_{vector_1}) \tag{9}$$

$$X^{'} = X + \beta(W_{y_i} - \underbrace{\frac{e^{W_{y_i}^T X} W_{y_i}}{\sum_{j=1}^{C} e^{W_j^T X} + e^{W_{virt}^T X}}}_{vector_2}) - \beta \underbrace{\frac{e^{W_{virt}^T X} W_{virt}}{\sum_{j=1}^{C} e^{W_j^T X} + e^{W_{virt}^T X}}}_{vector_3} \tag{10}$$

from the above Eq.9 and Eq.10, we consider the magnitude and direction of 'vector1', 'vector2' and 'vector3', and then the updated $X^{'}$ by the corresponding softmax and Virtual Softmax can be easily illustrated as in Fig.4.(c). It can be firstly observed that the norm of red arrow is smaller than that of blue arrow, showing a similar result with the analysis in Sec.5.1, i.e. Virtual Softmax will regularize the feature norm. Meanwhile, one can also observe another important phenomenon that the feature vector $X^{'}$ optimized by Virtual Softmax has a much smaller angle $\theta$ to the class anchor $W_{y_i}$ than original Softmax, well explaining the reason why the features learned by Virtual Softmax is much more compact and separable than original Softmax. In summary, by considering the collaborative effort of many terms, we could know the working principle of our Virtual Softmax better. **The Virtual Softmax not only provides regularization but more importantly, intensifies the discrimination property within the learned features.**

Although it is based on an approximate analysis, to some extent, it can give us an heuristic interpretation of Virtual Softmax why it is capable of encouraging the discrimination of features from a novel feature update perspective. And practically, without these assumptions the Virtual Softmax can indeed produce discriminative features, validated by the visualization in Sec. 5.3 and the experimental results in Sec. 6.

## 5.3 Visualization of Compactness and Separability

In order to highlight and emphasize that Virtual Softmax indeed encourages the discriminative feature learning, we provide a clear visualization of the learned features on MNIST dataset [17] in 2-D and 3-D space respectively, as shown in Fig.2. From it, one can observe that the learned features by our Virtual Softmax are much compact and well separated, with larger inter-class angular margin and tighter intra-class distribution than softmax, and Virtual Softmax can consistently improve the performances in both 2-D (99.2% *vs.* 98.91%) and 3-D (99.38% *vs.* 99.13%) cases. Furthermore, we also visualize the leaned features in k-D space (where $k > 3$) with the confusion matrix. The confusion matrix comparison between original softmax and our Virtual Softmax are shown in Fig.5. Specifically, we compute the included angle cosine (i.e. $\cos \theta$) between any two feature

| Layer | MNIST(for fig.2) | MNIST | SVHN | CIFAR10 | CIFAR100/100+ |
|---|---|---|---|---|---|
| Block1 | [5x5,32×t]x2,padding 2 | [3x3,32×t]x4 | [3x3,32×t]x5 | [3x3,32×t]x5 | [3x3,32×t]x5 |
| Pool1 | Max-Pooling | | | | |
| Block2 | [5x5,64×t]x2,padding 2 | [3x3,32×t]x4 | [3x3,64×t]x4 | [3x3,64×t]x4 | [3x3,64×t]x4 |
| Pool2 | Max-Pooling | | | | |
| Block3 | [5x5,128×t]x2,padding 2 | [3x3,32×t]x4 | [3x3,128×t]x4 | [3x3,128×t]x4 | [3x3,128×t]x4 |
| Pool3 | Max-Pooling | Max-Pooling | Max-Pooling | Max-Pooling | Ave-Pooling |
| Fully Connected | 2/3 | 64 | 64 | 64 | 512 |

Table 1: Model architectures for different benchmarks. [3x3,32×t]x4 denotes 4 cascaded convolutional layers with $32 \times t$ filters of size 3x3. The toy models are of $t = 1$. Max-Pooling is with $3 \times 3$ kernel and stride of 2.

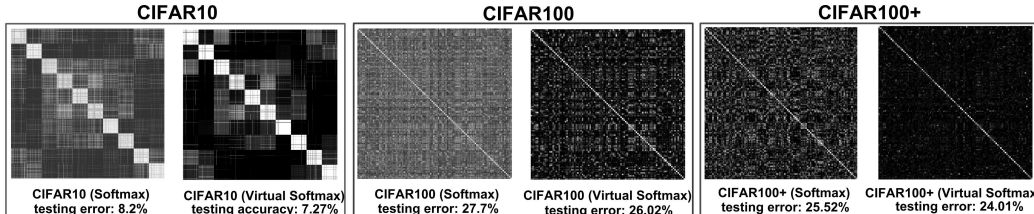

Figure 5: Cosine confusion matrixes on the test splits of CIFAR10/100/100+. On each dataset, both Softmax and Virtual Softmax use the same CNN architecture.

vectors which are extracted from the testing splits of CIFAR10/100/100+ [16] datasets. From Fig.5, it can be observed that, over the there datasets, the intra-class similarities are enhanced and the inter-class similarities are reduced when training with our Virtual Softmax. Moreover, our Virtual Softmax improves nearly $1\%$, $2\%$ and $1.5\%$ classification accuracies over the original softmax on CIFAR10/100/100+, respectively. To summarize, all these aforementioned experiments demonstrate that our Virtual Softmax indeed serves as an efficient algorithm which has a much stronger power of enlarging inter-class angular margin and compressing intra-class distribution than softmax and thus can significantly improve the recognition performances.

### 5.4 Relationship to Other Methods

There are a few research works concentrating on refining the Softmax objective, e.g. L-Softmax[22], A-Softmax[21] and Noisy Softmax[3]. Research works [22, 21] require to manually select a tractable constraint, and need a careful and annealing-like training procedure which is under human control. However, our Virtual-Softmax can be easily trained end-to-end without or with little human interventions, since the margin constraint is introduced automatically by the virtual class. Moreover, our method differs from L-Softmax and A-softmax in a heuristic idea that training deep models with the additional virtual class not only the given ones, ***which is the first work to extend the o-riginal Softmax to employ virtual class for discriminative feature learning and may inspire other researchers***. And it can also provide regularization for deep models implicitly. Method [3] aims to improve the generalization ability of DCNN by injecting noise into Softmax, which heads from a different perspective. In summary, our Virtual-Softmax comes from a clear and unusual motivation that injects a dynamic virtual class for enhancing features, which is different from the listed other methods[22, 21, 3], even though all the methods intend to learn better features.

## 6 Experiments and Results

We evaluate our Virtual Softmax on classification tasks and on face verification tasks. For fair comparison, we train with the same network for both Virtual Softmax and the baseline softmax.

**Small-Set Object Classification**: We follow [39, 22, 3] to devise our CNN models. Denote $t$ as the widening factor, which is used to multiply the basic number of filters in one convolutional layer, then our architecture configurations are listed in Table.1. For training, the initial learning rate is 0.1, and is divided by 10 at (20k, 27k) and (12k, 18k) in CIFAR100 and the other datasets respectively, and the corresponding total iterations are 30k and 20k.

**Fine-grained Object Classification**: We fine-tune some popular models pre-trained on ImageNet[26], including GooglenetV1[31] and GooglenetV2[32] , by replacing the last softmax layer with our Virtual Softmax. The learning rates are fixed to 0.0001 and 0.001 for the pre-trained layers and the randomly initialized layer respectively, and stop training at 30k iteration.

**Large-Set Object Classification**: Here, we use the network for CIFAR100 with t=7, the learning rate starts from 0.1 and is divided by 10 at 20k, 40k, 60k iterations. The maximal iteration is 70k.

**Face Verification**: We employ the published Resnet model in [21] to evaluate our Virtual Softmax. Start with a learning rate of 0.1, divide it by 10 at (30k, 50k) and stop training at 70k.

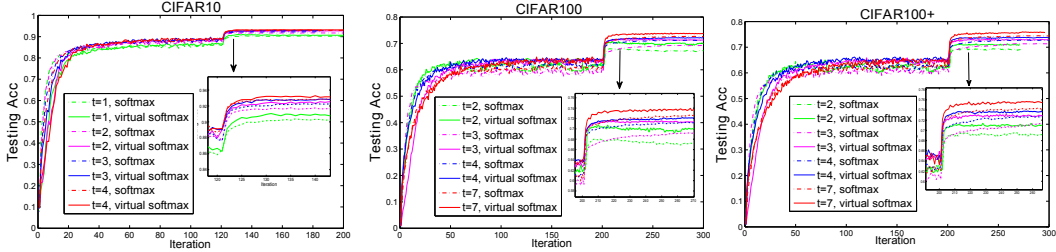

Figure 6: Testing Acc(%) on CIFAR10/100/100+ datasets.

**Compared Methods**: We set L-Softmax(LS)[22], Noisy-Softmax(NS)[3] and A-Softmax(AS)[21] as the compared methods and re-implement them with the same experimental configurations as us.

All of our experiments are implemented by Caffe[14]. The models are trained on one TitanX and we fill it with different batch sizes for different networks. For data preprocessing, we follow NS[3]. For testing, we use original softmax to classify the testing data in classification tasks and cosine distance to evaluate the performances in face verification tasks.

## 6.1 Ablation Study on Network Width

As listed in Table.1, our toy network on each dataset is with the widening factor of $t = 1$. We expand the toy models by setting $t = 1, 2, 3, 4$ and $t = 2, 3, 4, 7$ on CIFAR10 and CIFAR100 respectively, and the experimental results are listed in Table.2. From these results, for example on CIFAR100+, it can be observed that as the model width increasing (i.e. from $t = 2$ to $t = 7$) the recognition error rate of Virtual Softmax is diminishing, and our method achieves consistent performance gain over the original softmax when training with different networks, verifying the robustness of our method. Furthermore, when training with Virtual Softmax, the recognition error rates of wider models are consistently lower than that

| CIFAR10 | $t=1$ | $t=2$ | $t=3$ | $t=4$ |
|---|---|---|---|---|
| Softmax | 9.84 | 8.2 | 7.76 | 7.15 |
| Virtual Softmax | 8.95 | 7.27 | 7.04 | 6.68 |
| *Improvement* | *0.77* | *0.93* | *0.72* | *0.47* |
| **CIFAR100** | $t=2$ | $t=3$ | $t=4$ | $t=7$ |
| Softmax | 32.72 | 30.74 | 28.76 | 27.7 |
| Virtual Softmax | 29.84 | 28.4 | 27.81 | 26.02 |
| *Improvement* | *2.88* | *2.34* | *0.95* | *1.68* |
| **CIFAR100+** | $t=2$ | $t=3$ | $t=4$ | $t=7$ |
| Softmax | 30.44 | 28.59 | 27.21 | 25.52 |
| Virtual Softmax | 28.62 | 26.81 | 26.17 | 24.01 |
| *Improvement* | *1.82* | *1.78* | *1.04* | *1.51* |

Table 2: Recognition error rates(%) on CIFAR10/100 datasets. $t$ is the widening factor and + denotes data augmentation.

of thinner models across all these datasets, indicating that the Virutal Softmax does not easily suffer from over-fitting and supporting the analysis in Sec.5.1. And we plot the curves of testing Acc on CIFAR10/100/100+ as shown in Fig.6, one can observe that our Virtual Softmax can be easily optimized, with a similar convergence speed compared to the original softmax.

## 6.2 Evaluation on objecect datasets

**MNIST** [17], **SVHN** [23], **CIFAR** [16] are the popular used small-set object classification datasets, including different number of classes. From Table.3, it can be observed that on both **MNIST** and **SVHN** datasets the Virtual Softmax not only surpasses the original softmax using the same network (i.e. $0.28\%$ $vs.$ $0.35\%$ on MNIST, $1.93\%$ $vs.$ $2.11\%$ on SVHN) but also outperforms LS[22], NS[3] and AS[21], showing the effectiveness of our method. Moreover, we also report the experimental results on **CIFAR** dataset as in Table.4. Specifically, one can observe that our Virtual Softmax drastically improves nearly $0.6\%, 2\%, 1.5\%$ accuracies over the baseline softmax on CIFAR10/100/100+ respectively. Meanwhile, it outperforms all of the other methods on both CIFAR10/100 datasets, e.g. it surpasses both the RestNet-110[9] and Densenet-40[11] on CIFAR100+ which are much deeper and more complex than our architecture, and also surpasses the listed compared methods.

**CUB200** [35] is the popular fine-grained object classification set. The comparison results between other state-of-the-art research works and our Virtual Softmax are shown in Table.5, V1 and V2 denote the corresponding GoogleNet models. One can observe that the Virtual Softmax outperforms the baseline softmax over all the two pre-trained models, and surpasses all the compared methods LS[22], NS[3] and AS[21]. Additionally,

| Method | Top1 | Top5 |
|---|---|---|
| Softmax | 47.63 | 73.14 |
| NS*[3] | 47.96 | 73.25 |
| LS*[22] | 48.59 | 73.82 |
| AS*[21] | 48.66 | 73.57 |
| Virtual Softmax | *48.84* | *74.06* |

Table 7: Acc (%) on ImageNet32

training with only the Virtual Softmax, our final result is comparable to other remarkable works which exploit many assistant attention and alignment models, showing the superiority of our method.

| Method | MNIST(%) | SVHN(%) |
|---|---|---|
| Maxout [7] | 0.45 | 2.47 |
| DSN [19] | 0.39 | 1.92 |
| R-CNN [20] | 0.31 | 1.77 |
| WRN [39] | - | 1.85 |
| DisturbLabel [38] | 0.33 | 2.19 |
| Noisy Softmax [3] | 0.33 | - |
| L-Softmax [22] | 0.31 | - |
| Softmax | 0.35 | 2.11 |
| NS*[3] | 0.32 | 2.04 |
| LS*[22] | 0.30 | 2.01 |
| AS*[21] | 0.31 | 2.04 |
| *Virtual Softmax* | **0.28** | **1.93** |

Table 3: Recognition error rates on MNIST and SVHN. * denotes our reproducing.

| Method | CIFAR10(%) | CIFAR100(%) | CIFAR100+(%) |
|---|---|---|---|
| GenPool [18] | 7.62 | 32.37 | - |
| DisturbLabel [38] | 9.45 | 32.99 | 26.63 |
| Noisy Softmax [3] | 7.39 | 28.48 | - |
| L-Softmax [22] | 7.58 | 29.53 | - |
| ACU [13] | 7.12 | 27.47 | - |
| ResNet-110 [9] | - | - | 25.16 |
| Densenet-40 [11] | 7.00 | 27.55 | 24.42 |
| Softmax | 7.15 | 27.7 | 25.52 |
| NS*[3] | 6.91 | 26.33 | 25.20 |
| LS*[22] | 6.77 | 26.18 | 24.32 |
| AS*[21] | 6.83 | 26.09 | 24.11 |
| *Virtual Softmax* | **6.68** | **26.02** | **24.01** |

Table 4: Recognition error rates on CIFAR datasets. + denotes data augmentation. * denotes our reproducing.

| Method | CUB(%) |
|---|---|
| Pose Normalization [1] | 75.7 |
| Part-based RCNN [40] | 76.4 |
| VGG-BGLm [41] | 80.4 |
| PG Alignment [15] | 82.8 |
| Softmax (V1) | 73.5 |
| NS*[3](V1) | 74.8 |
| LS*[22](V1) | 76.5 |
| AS*[21](V1) | 75.2 |
| *Virtual Softmax* (V1) | **77.1** |
| Softmax (V2) | 77.2 |
| NS*[3](V2) | 77.9 |
| LS*[22](V2) | 80.5 |
| AS*[21](V2) | 80.2 |
| *Virtual Softmax* (V2) | **81.1** |

Table 5: Accuracy results on CUB200. * denotes our reproducing.

| Method | Models | LFW | SLLFW |
|---|---|---|---|
| DeepID2+ [30] | 25 | 99.47 | - |
| VGG [25] | 1 | 97.27 | 88.13 |
| Lightened CNN [37] | 1 | 98.13 | 91.22 |
| L-Softmax [22] | 1 | 98.71 | - |
| Center loss [36] | 1 | 99.05 | - |
| Noisy Softmax [3] | 1 | 99.18 | 94.50 |
| Normface [33] | 1 | 99.19 | - |
| A-Softmax [21] | 1 | 99.42 | - |
| Softmax | 1 | 99.10 | 94.59 |
| NS*[3] | 1 | 99.16 | 94.75 |
| LS*[22] | 1 | 99.37 | 95.58 |
| AS*[21]+Normface*[33] | 1 | **99.57** | **96.45** |
| *Virtual Softmax* | 1 | 99.46 | 95.85 |

Table 6: Verification results (%) on LFW/SLLFW. * denotes our reproducing.

**ImageNet32**[5]: is a downsampled version of large-scale dataset ImageNet [26], which contains exactly the same number of images as original dataset but with 32x32 size. The results are in Tab.7, one can observe that Virtual Softmax performs the best.

### 6.3 Evaluation on face datasets

**LFW** [12] is a popular face verification benchmark. **SLLFW** [6] generalizes the original protocol in LFW to a more difficult verification task, indicating that the images are all coming from LFW but the testing pairs are more knotty to verify. For data preprocessing, we follow the method in [3]. Then augment the training data by randomly mirroring. We adopt the public available Resnet model from A-Softmax[21]. Then we train this Resnet model with the cleaned subset of [8]. The final results are listed in Table.6. One can observe that our Virtual Softmax drastically improves the performances over the baseline softmax on both LFW and SLLFW datasets, e.g. from $99.10\%$ to $99.46\%$ and $94.59\%$ to $95.85\%$ , respectively. And it also outperforms both NS[3] and LS[22], showing the effectiveness of our method. Moreover, since the class number for training is very large, i.e. 67k, the optimizing procedure is difficult than that in object classification tasks, therefore, if the training phase can be eased by human guidance and control, the performance will be more better, e.g. AS*[21]+Normface*[33] achieve the best results when artificially and specifically choose the optimal margin constraints and feature scale for the current training set.

## 7 Conclusion

In this paper, we propose a novel but extremely simple method Virtual Softmax to enhance the discriminative property of learned features by encouraging larger angular margin between classes. It derives from a clear motivation and generalizes the optimization goal of the original softmax to a more rigorous one, i.e. zero limit $\theta_{y_i}$. Moreover, it also has heuristic interpretations from feature update and coupling decay perspectives. Extensive experiments on both object classification and face verification tasks validate that our Virtual Softmax can significantly outperforms the original softmax and indeed serves as an efficient feature-enhancing method.

**Acknowledgments**: This work was partially supported by the National Natural Science Foundation of China under Grant Nos. 61573068 and 61871052, Beijing Nova Program under Grant No. Z161100004916088, and sponsored by DiDi GAIA Research Collaboration Initiative.

## Footnotes

[1]Elucidated in [22]. Here for simplicity, assume that all $\|W_j\| = l$.

[2]N-dimensionality space complicates our analysis but has the similar mechanism as 2-D space.

[3]Assume that all the K classes are injected at the exact position among original C classes.

[4]Although the following discussion is based on this assumption, the actual uses of activation functions (like the piecewise linear function ReLU and PReLU) and the basis $b$ do not affect the final performances.

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
