[Reviews · NeurIPS 2018]

Reviewer 1



The paper proposes a simple technique for improved feature learning in convolutional neural networks. The technique consists of adding a “negative” virtual class to CNN training on classification tasks with the softmax loss function. The authors evaluate their approach on a range of computer vision datasets, (CIFAR10/100/100+, LFW, SLLFW, CUB200, ImageNet32) and find that it outperforms simple baselines on all of them, and outperforms more complicated state-of-the-art techniques on most of them. The authors also present an analysis from a few different standpoints as to why their method is effective. Strengths: - The technique proposed by the authors is extremely simple to implement (just a one line change in existing code would suffice, as far as I can tell). - Their technique performns well across a range of datasets, often outperforming more complicated approaches. Weaknesses: - I am not entirely convinced by any of the analysis presented in the paper. Section 5.1 seems to particularly lack rigor, as it ignores most of the terms when studying an objective being optimized. The authors acknowledge this, but have still included this analysis in the paper. The analysis in Section 5.2 also make a number of approximations (these are more reasonable, however). - The paper would benefit from some minor edits in English style/grammar. For example, in line 16-17: “While, many research works take” -> omit the “while”. In the end, I feel that the simplicity of the method, coupled with its strong performance on a large number of datasets merits an acceptance. However, the paper could be greatly improved with a better / more rigorous analysis of why it works well. Other comments: - Figure 2: What exact visualization technique is being used here? Would be good to mention that in the figure caption. - The authors criticise L-softmax and A-softmax for requiring an annealing-like training procedure, but this method itself has a very specific learning rate schedule for experiments on CIFAR and Face Verification.

Reviewer 2



The submission presents a new method named Virtual Softmax for deep learning based general classification task. Virtual Softmax aims at learning enlarged inter-class classification margin and compressed intra-class distribution via injecting a virtual class during training. The idea is simple, yet produces very promising results in multiple publicly available datasets in the object classification and the face verification. The paper is very well written and the experimental evaluation is solid. The proposed approach method is general and thus has very good application potential to different scenarios. The reviewer recommends an acceptance of the work, and does not have concerns regarding to the manuscript.

Reviewer 3



This paper proposes an interesting idea of using “Virtual Softmax” to improve the discriminative power of features learned in DNNs. The method works by injecting a dynamic virtual negative class into the original softmax, extending the softmax dimension from original C to C+1. The weight vector associated with the additional class is constructed to align with the current feature vector X_i, which effectively pushes the decision boundary of y_i to overlap with the class anchor weight vector. The author also provides some theoretical intuition w.r.t why adding the extra negative class helps the network to learn more separable and compact features. They verify the approaches experimentally on several tasks, and have shown superior performance. The proposed technique is easy to use and can be practically useful.